# Sublingual Immunotherapy: How Sublingual Allergen Administration Heals Allergic Diseases; Current Perspective about the Mode of Action

**DOI:** 10.3390/pathogens10020147

**Published:** 2021-02-02

**Authors:** Minoru Gotoh, Osamu Kaminuma

**Affiliations:** 1Department of Otorhinolaryngology, Nippon Medical School, Tokyo 113-8603, Japan; 2Laboratory of Allergy and Immunology, The Tokyo Metropolitan Institute of Medical Science, Tokyo 156-8506, Japan; 3Department of Disease Model, Research Institute of Radiation Biology and Medicine, Hiroshima University, Hiroshima 734-8553, Japan

**Keywords:** apoptosis, bitter taste receptor, CD4^+^ T cell, Japanese cedar pollinosis, microarray, proteomics, thrombospondin 1

## Abstract

Owing to the successful application of sublingual immunotherapy (SLIT), allergen immunotherapy (AIT) has become one of the leading treatments for allergic diseases. Similar to the case with other AITs, such as subcutaneous and oral immunotherapies, not only the alleviation of allergic symptoms, but also the curing of the diseases can be expected in patients undergoing SLIT. However, how and why such strong efficacy is obtained by SLIT, in which allergens are simply administered under the tongue, is not clearly known. Various potential mechanisms, including the induction of blocking antibodies, T cell tolerance, regulatory B and T cells, CD103^-^CD11b^+^ classical dendritic cells, and CD206^+^ macrophages, and the reduction of innate lymphoid cells, mast cells, and basophils, have been suggested. Recently, through a comparative analysis between high- and non-responder patients of SLIT, we have successfully proposed several novel mechanisms. Here, we introduce our recent findings and summarize the current understanding of the mechanisms underlying the strong efficacy of SLIT.

## 1. Introduction

The recent development of sublingual immunotherapy (SLIT) has brought a new era of the treatment of allergic diseases. This therapy has the following useful features: (1) the procedure is easy and involves simply holding an allergen solution or tablet under the tongue; (2) it has far fewer side effects than other immunotherapies (AITs), such as subcutaneous immunotherapy (SCIT) and oral immunotherapy (OIT), both of which carry a risk of inducing anaphylaxis; (3) unlike in the case of symptomatic treatments, one can expect not just downregulation of various symptoms, but also amelioration of allergic predisposition through SLIT; (4) it is associated with slower progression and less frequent onset of allergic diseases, that develop in organs other than the initial target organs. The requirement of long-term treatment for achieving satisfactory effectiveness and the existence of non-responder patients as described below are the few downsides of SLIT. From 2014 to 2018, at least five types of SLIT-based therapies have been approved in Japan. However, how SLIT exhibits such strong efficacy has not been fully understood so far. The amount of allergens employed in SLIT is much smaller than that required for inducing oral tolerance [1]. Unlike SCIT, allergens are not systemically administered to patients in SLIT. Although these features contribute to the safety of SLIT, the mechanisms behind its effectiveness are still unknown. In this review, we summarize the recently proposed potential mechanisms underlying SLIT, and introduce our novel findings obtained from a clinical study performed on patients with Japanese cedar pollinosis (JCP).

## 2. Antibody Responses

As the immune system responds to allergic materials, changes in serum and local levels of various immunoglobulins (Igs) are usually observed in patients who receive AIT, especially SCIT, in which allergens are systemically injected. IgE was identified as an Ig subclass responsible for allergen-induced skin reactions by Ishizaka et al. in 1966 [2]. Therefore, the effects of AIT have been investigated, especially with respect to serum IgE dynamics, for more than half a century. An increase, rather than a decrease, in serum allergen-specific and total IgE levels is often seen within the initial weeks or months of SCIT [3]. A decrease in serum IgE levels after long-term treatment, and the prevention of seasonal increase in IgE levels during the pollen season, have been reported in some studies [4,5,6]. However, these changes are not significant enough in most cases of SLIT [7,8,9]. 

In contrast to IgE, dynamic changes in the levels of other abundant classes of Igs, such as IgG and IgA, are often seen following AIT. Increases in serum IgG1, IgG4, and IgA levels in some cases were induced by SCIT in association with the improvement of allergic symptoms [6,10]. These Igs have the potential to inhibit IgE-dependent responses. IgE-mediated augmentation of allergen presentation from B cells to T cells was diminished on adding an IgG/IgA-containing serum fraction of allergic patients who received AIT [11], and was enhanced on adding an IgG4-depleting serum fraction [12]. IgE-dependent basophil degranulation was inhibited by IgG4 purified from AIT-treated patient sera [13]. It was shown that the allergen specificity, but not class or subclass of Igs, was critical in blocking IgE-mediated responses [14].

A weak but significant increase in allergen-specific IgG level, especially IgG4, by SLIT, has been reported in some clinical studies [7,8,15,16] but not others [9,17]. A decrease in the IgE/IgG4 ratio was observed following SLIT, which was correlated with the suppression of allergen-induced skin reactions and/or clinical symptoms in some studies [18,19] but not all [9,20]. An increase in serum IgA level has hardly been seen in SLIT [17]. In mouse models of bronchial asthma comparing the efficacy of SCIT and SLIT, alleviation of bronchial hyperresponsiveness was seen in both treatments, although upregulation of IgG1 was induced by SCIT alone [21]. Shirinbak et al. demonstrated that SCIT suppressed allergen-induced airway inflammation in immunized mice, independent of systemic IgG and IgA responses [22]. Considering the strong efficacy of SLIT, which is close to SCIT, and its relatively weak effect on Igs, the principal pharmacological mechanisms underlying SLIT may not rely on antibody responses.

## 3. B Cell Responses

Since B cells and B cell-derived plasma cells are predominant producers of Igs, B cells are a candidate target cell type for improving the effectiveness of AIT. SCIT targeting B cells with B cell epitope-based vaccine, that did not induce IgE-dependent responses, significantly alleviated nasal symptoms of grass pollen allergy patients [23]. An increase in allergen-specific memory type peripheral B cell numbers expressing IgA or IgG4 was induced by SCIT [24], although this has not been reported in SLIT studies. Furthermore, Hoof et al. recently reported that memory B cells were responsible for AIT-mediated rapid elevation of serum IgE levels, by inducing Ig class switching [25].

In addition to Ig-producing B cells, regulatory B (Breg) cells are implicated in the efficacy of AIT. Breg cells are characterized by the expression of CD19, CD25, and CD71 as cell surface molecules, and the production of interleukin (IL)-10, a regulatory cytokine [26,27]. Breg cells produce less IgE, and suppress T cell and dendritic cell (DC) activities [28]. Although a decrease in peripheral Breg cell number was observed in patients with allergic rhinitis (AR) and bronchial asthma [27,29], the cell numbers recovered by receiving AIT [30]. The amelioration of allergen-induced airway inflammation in a murine model of asthma was achieved by the adoptive transfer of IL-10-producing Breg cells [31], suggesting that the differentiation of Breg cells may contribute to the suppression of symptoms in allergic patients receiving AIT. Whether Breg cells are induced by SLIT as well as the mechanisms of Breg cell differentiation remain to be investigated. 

## 4. T Cell Responses

T cell-mediated immune responses are affected by AIT. The allergen-induced proliferative response of peripheral T cells was downregulated following SCIT [32]. The capacity of CD4^+^ T cells to produce helper T (Th) 2 cytokines such as IL-4 and IL-5 was also diminished [32,33,34,35], whereas increased [33,35], unchanged [32,36], and decreased [37] interferon (IFN)-γ production was reported by different studies. The systemic effect of SLIT is weaker than that of SCIT [38], as in the case of antibody responses, although the downregulation of proliferative response as well as Th2 cytokine production has been observed in several clinical studies of SLIT [39,40,41]. Since Th2 cytokines, such as IL-4 and IL-5, are crucial for IgE production and eosinophilic inflammation, respectively [42,43], downregulation of Th2 cell subset is likely to be directly and indirectly associated with alleviation of allergic symptoms. Li et al. demonstrated that a decrease in serum IL-17, but not IL-5, IL-13, or IFN-γ levels, was observed after a 2-year SLIT in correlation with the improvement of symptom scores [36]. Whether these weak changes in serum cytokine levels and cytokine-producing properties of peripheral T cells are crucial or negligible in the efficacy of SLIT, requires additional investigation. 

The induction of Treg cells is one of the most likely mechanisms behind the efficacy of AIT. In addition to various studies of AITs employing other allergen administration routes [44,45], phenotypes related to Treg cell induction are induced by SLIT. Phenotypic transformation of circulating follicular helper T (Tfh) cells into Treg cells was observed following SCIT and SLIT [46]. Since Tfh cells help B cells to produce IgE as well as other Ig classes and subclasses through the production of IL-21 [47], its reduction may augment the efficacy of AIT by affecting antibody responses. The augmented production of IL-10 was observed in peripheral T cells of AR patients following SLIT [40,48]. Consistently, Treg cells expressing CD25, Forkhead box P3 (FoxP3), a transcription factor responsible for Treg cell differentiation, and IL-10, were peripherally induced by SLIT, in association with the downregulation of allergen-induced T cell proliferation [41]. The proliferative response was recovered by depleting CD25^+^ cells or through anti-IL-10 antibody treatment [41]. IL-10 directly downregulates and upregulates the production of IgE and IgG4, respectively [49], suggesting that AIT-induced Treg cell induction is also related to antibody responses. However, changes in the production of TGF-β, a characteristic Treg-derived cytokine, were not often seen in clinical studies of SLIT [40,41]. IL-35 was recently identified as another Treg-derived cytokine. The proliferation and cytokine production of allergen-specific Th2 cells, and IgE production by B cells, were inhibited by IL-35 [50]. Increases in the number of peripheral blood IL-35-producing Treg cells, and their capacity to produce IL-35, were observed in patients with AR following SLIT [50]. The induction of CD8^+^ suppressor T cells was hardly seen in SLIT [32], although upregulation of CD8^+^CD25^+^CD137^+^ Treg cells by SCIT was reported [51]. Since the subdivision of Treg cells has been proceeding [52], in future, other Treg subsets may also be found to influence the efficacy of SLIT. 

## 5. Contribution of Allergen-Presenting Cells (APC)

Allergens topically administered under the tongue in SLIT are hardly absorbed through the oral mucosa [53]. Nagai et al., demonstrated in a mouse study that sublingually administered allergens were transported across sublingual ductal epithelial cells to APCs located in the oral mucosa [54]. Langerhans cells, macrophages, and two types of classical DCs (cDCs), with CD103^+^CD11b^−^ and CD103^−^CD11b^+^ phenotypes, have been identified as major oral APCs [55]. CD103^−^CD11b^+^ cDCs produce retinoic acid (RA) following migration to draining lymph nodes [56], although the same activity was observed in CD103^+^CD11b^−^cDCs located in other tissues [57,58]. Since RA is essential for Treg cell differentiation, CD103^−^CD11b^+^ cDCs may specifically contribute to the efficacy of SLIT through Treg cell induction. Recently, Yang et al. identified a new CD206^+^ macrophage subset in the sublingual mucosa [59]. Increased CD206^+^ macrophage numbers in response to sublingual allergen application alleviated the allergen presentation property of cDCs by producing IL-10 [59]. Since tolerogenic CD103^−^CD11b^+^ cDCs and the opposite phenotype cDCs are co-localized in the oral mucosa, the augmentation of SLIT efficacy may be achieved by developing a means to selectively deliver allergens to CD103^−^CD11b^+^ cDCs.

## 6. New Mechanisms and Immunological Markers of SLIT Efficacy

### 6.1. Induction of Mast Cell Degranulation Inhibitor

To elucidate the mechanisms underlying the strong efficacy of SLIT, despite the local application of a lower amount of allergen, we performed a series of clinical studies. Approximately 200 adult patients with JCP were treated by SLIT with a cedar pollen allergen for 2 years. Serum sampling was performed before and after the SLIT. Furthermore, we utilized the remarkable feature of SLIT, that it is often ineffective for approximately 30% of patients, even after treatment for several years [60,61]. Thus, the efficacy of SLIT was evaluated not through comparison with placebo groups, but through comparison between groups of patients showing obvious improvement of symptoms (high-responders; HRs) and unchanged or exacerbated conditions (non-responders; NRs), upon administration of active allergen. Based on this setting, systematic comparisons between before (pre) and after (post) the SLIT and between HRs and NRs were performed for cellular, serum, and genetic parameters. 

The dynamics of serum Igs were not consistent, at least in part, with previous theories as described above. A significant increase, rather than a decrease, in allergen-specific serum IgE level, was observed in both HRs and NRs after SLIT. An increase in IgG4 was also observed, but it was significant only in NRs [62], suggesting that the efficacy of SLIT is not mainly due to the effect on those Igs. However, it is believed that many symptoms of JCP, such as sneezing, rhinorrhea, and nasal congestion, are induced by chemical mediators including histamine and leukotrienes derived from mast cells or basophils [63,64]. Therefore, whether mast cell degranulation was affected by SLIT despite serum IgE elevation in our study was the next concern. Remarkably, IgE-dependent degranulation of CD34^+^ cell-derived mast cells was strongly suppressed by the addition of post-SLIT sera of HRs, but not NRs [65]. The downregulation of allergen-induced mast cell degranulation by AIT was consistently reproduced in a mouse model [66]. A proteomic analysis revealed that thrombospondin (THRS)-1 predominantly induced in HR sera after SLIT alleviated mast cell degranulation (Figure 1) [65]. Yang et al. reported that THRS-1 derived from B cells induced the generation of tolerogenic cDCs [67]. Confirming the pharmacological effect of THRS-1, for example, in animal studies, and elucidating the mechanisms underlying SLIT-mediated upregulation are objectives worth pursuing. 

### 6.2. Correlation among Cytokines

As the levels of various serum cytokines were different between pre- and post-SLIT, the difference between HR and NR was hardly observed. Among the 50 cytokines examined, the concentration of IL-12 was higher in HR, although it was only significant before SLIT. However, the hierarchical clustering analysis of these cytokines revealed that typical Th1/Th2 cytokines including IL-4, IL-5, IL-13, and IFN-γ showed a higher correlation in HR than NR (Figure 2) [62]. Besides the Th1/Th2 theory originally proposed by Mossman et al. in 1986 [68], many other T cell subsets and other cells have been implicated in the pathogenesis of allergic diseases [69]. The endotypes of individual patients may be a crucial factor in deciding the responsiveness to SLIT. 

### 6.3. Innate Lymphoid Cells (ILCs)

Among other cells in allergic diseases, several types of ILCs that do not show allergen specificity but produce large amounts of cytokines, have recently been identified. From their characteristic cytokines and transcription factors, type 1 ILCs (ILC1s), ILC2s, and ILC3s resemble Th1, Th2, and Th17/Th22 cells, respectively [70]. An increase in peripheral blood ILC2/ILC1 ratio was observed in AR patients [71]. IL-5 and IL-13 are produced by peripheral blood mononuclear cells of asthmatic patients upon stimulation with IL-25 or IL-33, a typical ILC stimulator, although it was only weakly observed in AR patients [72]. Lao-Araya reported that the seasonal increase in peripheral ILC2s in AR patients was suppressed by SCIT [73]. Although ILC3 has been implicated in immune tolerance induction by AIT [74], downregulation of ILC3 in AR patients undergoing SCIT was also reported [75]. Whether changes in the phenotypes of ILCs are involved in SLIT efficacy needs to be evaluated.

### 6.4. Taste Receptors in CD4^+^ T Cells 

Regardless of the differences in Th1/Th2 cytokines, the peripheral CD4^+^ T cell population was unaffected by SLIT in our study [62]. However, genome-wide transcriptome and nucleotide variation analysis revealed the potential genetic predisposition responsible for SLIT efficacy. Compared with NRs, elevated expression of several bitter taste receptors in CD4^+^ T cells associated with the increment of genetic copy number, was seen in HRs. In fact, stimulation-induced IL-4 expression in T cells was upregulated by bitter taste receptor agonists [76]. Since a substantial number of CD4^+^ T cells are located in taste buds of the tongue [77], stimulation of these T cells through bitter taste receptors may be involved in the effectiveness of SLIT.

### 6.5. Apoptosis of CD4^+^ T Cells and Basophils

Multivariate pathway analysis of genome-wide transcriptome data suggests other possible mechanisms of SLIT. In the comparison between pre- and post-SLIT HRs or post-SLIT HRs and NRs, the involvement of an apoptosis pathway was indicated not only in CD4^+^ T cells, but also in basophils, in context of SLIT efficacy [78]. Consistently, Matsumoto et al. reported that stimulation through Fas (CD95) expressed on basophils induced their apoptosis [79]. In addition to mast cells, basophils are important suppliers of histamine and other chemical mediators [63,64]. The reduction of mast cells and/or basophils in nasal swabs has been observed in a recent clinical study of SLIT in JPC patients [80]. The involvement of basophils in the pathogenesis of allergic inflammation has been debated in both animal and clinical studies [64,81]. Elucidating the cascades involved in SLIT-mediated apoptosis induction in basophils as well as CD4^+^ T cells, may strengthen their contribution to the effectiveness mechanisms. 

Remarkably, networking molecules related to the Treg cell inducing pathway did not appear in our multivariate analysis. Although Treg cells have been recognized as a key player for immunosuppression as described above, we previously demonstrated that the suppressive effect of OIT using transgenic rice expressing cedar pollen allergens in a murine model of asthma was mainly caused by the induction of the apoptosis/anergy of allergen-reactive CD4^+^ T cells, but not by the induction of Treg cells [82]. Detailed and comparative investigation of Treg cell- and apoptosis-inducing pathways in the mechanisms underlying SLIT efficacy will be required. 

## 7. Conclusions

In addition to revealing new mechanisms and biomarkers, our recent examinations have elucidated the clinical application of SLIT to treat AR in Japan. Through a double-blind, randomized, comparative study demonstrating high efficacy and safety [83], the first SLIT against JCP with crude cedar pollen extract was launched in 2014. Then, the application of a higher dose with a change in the dosage form from the extract to the tablet was achieved through the 3-year treatment, plus a 2-year follow-up study [84]. Based on our recent investigation [85], SLIT containing a combination of house dust mite tablets and Japanese cedar pollen tablets has been approved. Currently, SLIT is proactively used for AR patients, from 5-year-old children to adults, and is under trial for application in other allergic diseases in Japan.

As per the findings of various basic and clinical studies of SLIT, including our recent investigations, and other AITs as described above, multiple mechanisms seem to be involved in the strong efficacy of SLIT against allergic diseases (Figure 3). Focusing on the individual mechanisms, as well as enhancing the grovel efficacy, are potential avenues for further encouraging the use of SLIT. For example, the development of supportive treatments exploiting the individual mechanisms may strengthen the efficacy and shorten the treatment period of SLIT. To further improve the efficacy of this therapy, the identification of biomarkers that enable us to distinguish between NRs before the start of SLIT, as well as the elucidation of a key mechanism that determines the unresponsiveness of NRs against SLIT, needs to be done.

## Figures and Tables

**Figure 1 pathogens-10-00147-f001:**
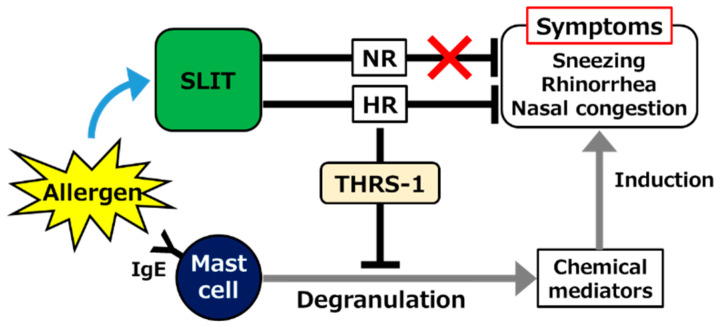
Schematic showing the contribution of thrombospondin (THRS)-1 as a mast cell degranulation inhibitor induced in sera of high-responder (HR) but not non-responder (NR) patients with Japanese cedar pollinosis (JCP), following sublingual immunotherapy (SLIT). IgE; immunoglobulin E.

**Figure 2 pathogens-10-00147-f002:**
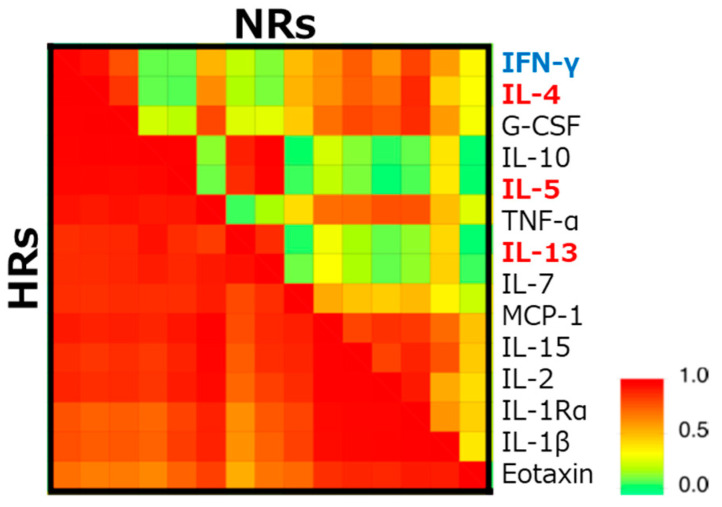
Difference in the correlation coefficients of serum cytokines between high-responders (HRs) and non-responders (NRs). The correlation coefficients between concentrations of cytokines in sera of HR and NR before sublingual immunotherapy (SLIT), are examined following hierarchical clustering by an unweighted pair–group method with the arithmetic mean. Blue and red fonts indicate representative helper T (Th) 1 and Th2 cytokines, respectively [62]. IFN; interferon, IL; interleukin, G-CSF; granulocyte-colony stimulating factor, TNF; tumor necrosis factor, MCP; monocyte chemotactic protein.

**Figure 3 pathogens-10-00147-f003:**
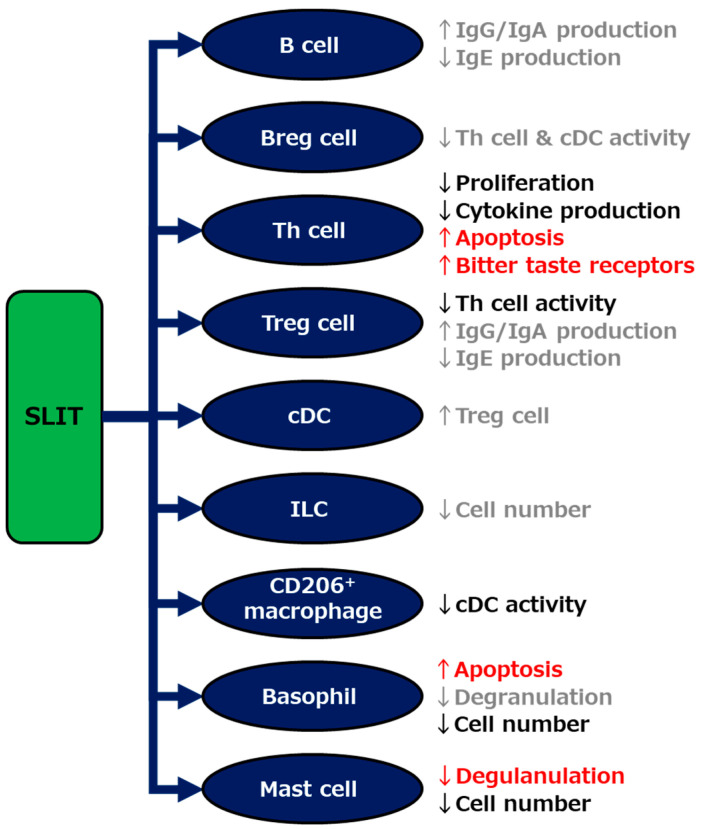
Schematic showing mechanisms underlying the efficacy of sublingual immunotherapy (SLIT). The mechanisms indicated by our studies and others are described in red and black, respectively, and those indirectly suggested elsewhere are in gray. Breg; regulatory B, Th; helper T, Treg; regulatory T, cDC; classical dendritic cell, ILC; innate lymphoid cell.

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
