# Peer review of "Sublingual Immunotherapy: How Sublingual Allergen Administration Heals Allergic Diseases; Current Perspective about the Mode of Action"

_pathogens, 2021, doi:10.3390/pathogens10020147_

Round 1

Reviewer 1 Report

The review manuscript "Sublingual immunotherapy: how sublingual allergen administration heals allergic diseases, current perspective about the mode of action" is a comprehensive compendium of what it is currently known about SLIT. The references are appropriate.

However, the structure of the review is, in my opinion, not very straightforward. The authors use the 5 first headings to introduce SLIT and compare it with SCIT (1), and revise what is known regarding antibody (2), B cells (3), T cells (4) and APC (5) responses in SLIT. However, the 6th heading (new mechanisms of SLIT efficacy) is a mix of information about ILCs, cytokines and genes without any structure or subheadings.

I strongly recommend rewriting this part of the review.

Moreover, there are typos in the figures:

Figure 1: Deglanulation

Figure 3: Degulanulation

Please, revise and ammend typos.

Finally, the review would benefit from a native English speaker proofreading.

Reviewer 2 Report

The manuscript submitted by Gotoh et al., provides a nice review about sublingual immunotherapy (SLIT), and the role of different cells and mediators of the immune system to orchestrate clinical benefit following SLIT. Yet, many mechanisms involved are still unclear.

The revision is well written, and of interest for a broad number of researchers and clinicians working in allergy/asthma. A few considerations must be addressed before its acceptance:

  • Page 1 lines 38-39, “The requirement of long-term for achieving satisfactory effectives is the only downside of SLIT” should be softened, it is not the only one, the efficacy, for instance, is not always as good as expected.
  • Page 1, lines 79-80, “Considering the strong efficacy of SLIT, which is comparable to SCIT, and it relatively…”. In general, SLIT is considered to be safer than SCIT, but its efficacy and effectiveness is still in most cases lower. This sentence should be rephrased accordingly.
  • Page 4, lines 148-149, “CD103-CD11b+ cDCs produce retinoic acid (RA) following migration to draining lymph nodes [56]. Since RA is essential for Treg cell differentiation, CD103-CD11b+ cDCs may specifically contribute to the efficacy of SLIT through Treg cell induction.” Other studies have suggested that CD103+CD11b- rather than CD103-CD11b+ are responsible for Treg induction, homeostasis and RA production. This should be acknowledged and briefly discussed in this section.
  • Page 4, line 158 “New mechanism of SLIT efficacy”, this heading should be replaced by a more appropriate one, such as “Immunological markers of SLIT”. In the text that follows, it is explained the difference in the expression/levels of different cytokines and other markers in high-responders vs non-responders following SLIT treatment, but an underlying mechanism is not really provided/demonstrated.
  • Page 4, Figure 1, “Deglanulation” should be replaced by “degranulation”.
  • Page 6, figure 3 “Degulanulation” should be replaced by “degranulation”. In this figure it is indicated in grey that ILCs decrease cell numbers. This might be the case for ILC2; however, there is a number of studies that suggest that ILC3 are increased in AIT (DOI: 10.1016/j.iac.2019.09.009). This should be somehow indicated in the figure.
  • Page 7, line 253, “In addition to revealing several new mechanisms….” Should be replaced by “In addition to revealing new biomarkers”.
